# Simulated Guidance in Interpreting Nano-Patterned Co_70_Fe_30_ Film Imaging with Differential Phase Contrast

**DOI:** 10.3390/nano14010116

**Published:** 2024-01-03

**Authors:** Björn Büker, Daniela Ramermann, Pierre-M. Piel, Judith Bünte, Inga Ennen, Andreas Hütten

**Affiliations:** Faculty of Physics, Bielefeld University, Universitaetsstraße 25, 33615 Bielefeld, Germany; bueker.bjoern@gmail.com (B.B.); daniela.ramermann@cec.mpg.de (D.R.); ppiel@uni-muenster.de (P.-M.P.); judith.buente@uni-bielefeld.de (J.B.); ennen@physik.uni-bielefeld.de (I.E.)

**Keywords:** differential phase contrast (DPC) magnetic imaging, nano-patterned thin-film membranes, focused ion beam (FIB) milling, micromagnetic simulations

## Abstract

Our paper introduces a simulation-based framework designed to interpret differential phase contrast (DPC) magnetic imaging within the transmission electron microscope (TEM). We investigate patterned magnetic membranes, particularly focusing on nano-patterned Co_70_Fe_30_ thin-film membranes fabricated via focused ion beam (FIB) milling. Our direct magnetic imaging reveals regular magnetic domain patterns in these carefully prepared systems. Notably, the observed magnetic structure aligns precisely with micromagnetic simulations based on the dimensions of the underlying nanostructures. This agreement emphasizes the usefulness of micromagnetic simulations, not only for the interpretation of DPC data, but also for the prediction of possible microstructures in magnetic sensor systems with nano-patterns.

## 1. Introduction

In recent years, interest in nanostructured materials, so-called “metamaterials”, has increased significantly [1,2]. For the investigation of magnetic materials on the nanoscale, techniques that can directly image the magnetic structure of a sample are especially highly desirable. Current methods, such as the magneto-optical Kerr effect (MOKE), have limited resolution [3] or require a synchrotron-based setup such as X-ray photoemission electron microscopy (XPEEM) [4]. Although other techniques with potentially high resolution such as magnetic force microscopy (MFM) [5] provide information about the magnetic domain structure in a sample, information about the orientation of these domains relative to each other is lost.

In this work, we present a simulation-based workflow for the interpretation of magnetic differential phase contrast (DPC) imaging in TEM to study the magnetic structure of nano-patterned Co_70_Fe_30_ membranes [6,7,8] as a model system. The DPC technique was developed more than 10 years ago. However, most of the previous micromagnetic studies rely on the MFM method, whereas the DPC technique is successfully used in this work. With this technique, we can study both the electronic and magnetic properties of a sample by detecting the resulting small deflections of the transmitted electron beam with a segmented detector [9]. To support the experiment, micromagnetic simulations provide an important context that helps to identify artifacts in the measurement and predict potentially interesting sample structures. Recent advances in DPC microscopy have made it possible to study the electronic structure of a material with atomic resolution [10,11]. However, due to the limitations in producing a suitable sample for this technique, it is difficult to produce a sufficiently large sample for the study of magnetic domains. The morphology of the model samples used was inspired by work on artificial spin ice, with the aim of creating complex magnetic configurations that challenge the capabilities of the DPC technique [12,13,14].

To this end, large magnetic thin-film membranes were fabricated specifically for use with DPC in TEM with focused ion beam (FIB) milling. We will show that these nano-patterned model samples have an interesting magnetic domain structure by using magnetic imaging with DPC together with a simulation framework based on micromagnetic calculations to help interpret the experimental data. These simulations not only agree very well with the measured, data but are also consistent over a wide range of possible sample structures. This direct correlation between simulation and experiment can be used in the future to design architectures of nano-patterned systems at the simulation level, which then come very close to the experiment without having to go through many fabrication cycles with subsequent characterization during fabrication.

The overarching goal of future investigations based on this combination of DPC and micromagnetic interpretation is therefore the development of an in-situ magnetometer in the TEM that enables quantitative imaging analysis of magnetic switching processes. The use of TEM heating holders would then make the possible changes in switching behavior at higher ambient temperatures and the determination of temperature-dependent magnetic properties accessible. The design of magnetoresistive components based on the architecture of nanostructured multilayers, for example, is thus raised to a higher level.

## 2. Experimental and Calculation Methods

The TEM specimens were fabricated on SiN disk substrates in the size of a TEM grid with a 50 µm × 50 µm window covered by a 10 nm thick SiN membrane in the center. To create sufficient beam displacement for the DPC measurement, 60 nm thick Co_70_Fe_30_ layers were deposited from an alloy target onto the disks using DC magnetron sputter deposition in a customized Leybold T11600 magnetron sputtering system from 3SC Leybold Germany. This system supports up to eight targets in the sputtering chamber, of which seven are 4″ in diameter and one is a 2″ target. In addition, a vapor deposition system is available in the center of the chamber. The deposition processes can be automated via the machine’s programmable control interface to ensure consistent results across multiple samples. Uniform growth of the Co_70_Fe_30_ film is ensured by the first deposition of a 5 nm thick Py (Ni_81_Fe_19_) seed layer. A 3 nm thick Ru top layer protects the sample from corrosion. The base pressure of the sputtering system was p_base_ ≤ 10^−7^ mbar at a constant Ar flow of 20 sccm with a working pressure of p_work_ = 2.3·10^−3^ mbar. Proven standard values of 29 W for the 2″ Py target and 60 W each for the 4″ Ru and Co_70_Fe_30_ targets were selected as sputtering powers. The choice of Co_70_Fe_30_ as the magnetic layer can be justified as follows: firstly, the magnetic material properties of Co_70_Fe_30_ are very well studied in our laboratory, and secondly, the optimal deposition conditions for homogeneous alloy layers of the same thickness are well known.

Since the mechanical stress on the sample during lithography easily breaks the SiN membrane, samples were structured in a dual beam focused ion beam (FIB) microscope using a Ga+ ion beam at an acceleration voltage of 30 kV with a relatively low beam current of 9.7 pA for high resolution patterning. This process can be automated using scripts for regular, easily repeatable patterns on a grid. In principle it is possible to create arbitrary hole shapes in this manner; however, as a proof-of-concept, simple shapes such as circular and rectangular holes on a rectangular grid were chosen. With the given beam parameters, the minimum feature size lies well below 100 nm, but for the purpose of DPC magnetic imaging, feature sizes and distances of several hundred nanometers were chosen so that domain wall sizes are not a constraining factor.

Differential phase contrast measurements on the samples were performed in a JEOL JEM-ARM200F transmission electron microscope at 200 kV. Before each DPC measurement, the samples are magnetically saturated in the in-plane direction. By tilting the sample holder around the gonio axis, which is aligned perpendicular to the incident electron beam and the magnetic field of the objective lens, an in-plane magnetic component is generated in the sample that is sufficient to completely saturate the soft magnetic layers. The magnetic field of the objective lens is controlled via the free lens control so that the in-plane component in the sample is approximately 1600 mT at a tilt angle of 25°. This ensures that the in-plane component saturates the magnetization of the sample and thus induces a preferred direction for the magnetic domains near the remanence.

Since a field-free environment around the sample is important to avoid artifacts, the objective lens is switched off and the sample holder is tilted back to 0° before imaging. Absolute beam deflection and angle are recorded by an 8-fold segmented DF-STEM detector and can be mapped to a color wheel to visualize the orientation of magnetic domains to one another [6,8]. As the orientation of the detector segments does not necessarily align with the pattern on the sample, the data from the two deflection channels recorded by the detector can be transformed to match the lattice axes of the pattern without loss of information. Data can also be normalized using empty areas such as the holes, since the beam deflection there should be close to zero. However, it should be noted that the color representation of DPC images in general depends on several different factors such as the scanning direction of the beam, the orientation of the sample, and the relative angle of sample and electron beam. Additionally, for a large field of view, a deflection—i.e., color—gradient is superimposed to the signal, since the beam deflection needed to scan a large area also influences the recorded beam deflection. Although it is possible to quantitatively account for these factors, considerable effort and knowledge of the sample are required to program an image correction routine for this application, which goes beyond the scope of this work. Instead, as the absolute value of the magnetic induction is hard to discern, the discussion will focus on the relative orientation of the observed magnetic domains, which is always imaged accurately. Additionally, each sample area is imaged a second time after saturating the magnetization of the sample in the opposite direction to confirm the magnetic origin of the beam deflection.

Furthermore, the samples were imaged by a JEOL JEM-2200FS using Lorentz microscopy in the Fresnel mode, which helps to identify domain walls [15]. Together with high-resolution scanning electron microscope (SEM) images after sample preparation, it can be ensured that the observed structures in the DPC images are indeed of magnetic origin and not a result of defects in the material.

Micromagnetic simulations have seen significant advancements, especially through prominent software packages such as Mumax [16] and OOMMF [17]. These packages continue to be pivotal in studying magnetic behaviors at nanometer scales, aiding research in diverse fields from data storage to spintronics. Mumax, known for its efficiency in handling large-scale simulations with GPU acceleration, has undergone updates improving its performance and versatility, allowing researchers to model complex systems more accurately. On the other hand, OOMMF remains a robust choice, because of its open-source nature and adaptable framework, enabling users to customize simulations to specific experimental conditions. Both packages contribute significantly to the field, with Mumax excelling in speed and large-scale simulations, while OOMMF’s flexibility and accessibility support a wide range of research endeavors.

Nevertheless, motivated by our many years of experience with the micromagnetic simulation program package MicroMagus 7.1ext [18], all micromagnetically simulated nanostructured thin films shown here were created with this program package. With this program, it is possible to perform quasi-static simulations to determine the equilibrium magnetization configuration of a system in each external field [19]. Consequently, it is possible to obtain a set of magnetization configurations for a complete magnetic field loop using a series of individual calculations. The program additionally produces text output files containing a list of all energy contributions for all simulation steps, as well as the total magnetic moments for all applied fields and simulation steps, which allows for the creation of hysteresis loop. In the case of multilayer systems, the program requires a geometric input detailing the layer structure and shape, a set of magnetic parameters such as saturation magnetization and an anisotropy constant, which can be found in the literature, and a set of external field values. Moreover, the program can work with all available CPU threads at once. Additionally, periodic boundary conditions were used to avoid artifacts and more accurately represent the true size of the samples used in the experiment. For polycrystalline samples, it is possible to additionally supply a noise pattern with certain dimensions to imitate a random granular structure with an average grain size. Detailed information on the involved mathematics can be found on the official program website, [18], or in [19] by the same author.

With the sample parameters and material constants given in Table 1, the code outputs a magnetic configuration for a given external field. Although there are many well-comparable magnetic data for bulk materials, the range of these data for ultrathin films is very wide. We therefore created a database that matches the micromagnetic simulations quite well with the measured properties, considering the substrate used. The data listed in Table 1 are taken from this database. To visualize the switching dynamics in the sample, a complete reversal loop between −1000 mT and 1000 mT was calculated. Around µ_0_H_ext_ = 0, the step size for the external field is shrunk appropriately to visualize the switching process of the film. The color representation for these simulations was chosen so that similarly oriented domains are the same color in both simulation and experimental data. It is important to acknowledge that the layer thicknesses used in the simulation are 0.75 nm for Py and 9 nm for Co_70_Fe_30_, which are smaller than in the experiment but keep the relative thickness of the layers the same. Therefore, in the simulations, the layers were made to be in the order of the exchange length so that these can be modeled using single cells through each layer’s thickness.

## 3. Experimental Results and Discussion

The FIB-milled Co_70_Fe_30_ membranes shown by the SEM images in Figure 1 exhibit high mechanical stability and do not deteriorate even in the 200 kV electron beam of the TEM. Furthermore, the images clearly show regular structures having a size of several hundred nanometers, which are ideal for imaging using DPC. The rectangular pattern shown in Figure 1a) consists of 560 nm × 310 nm holes with an edge-to-edge distance between the holes of 430 nm in the x-direction and 360 nm in the y-direction. For the grid of circular holes (Figure 1b), a diameter of 500 nm and a distance of roughly 480 nm were chosen. Smaller edge-to-edge distances of isolated features down to 150 nm showed no impact on the mechanical stability of the membrane.

To first confirm the magnetic origin of the differential phase contrast signal, a large isolated square defect of 2 µm × 2 µm was observed in the TEM. Figure 2a shows the DPC image of the sample at a remanence 0° tilt relative to the incident beam after being saturated by the objective lens at a sample tilt of 25°. The square was imaged again, as shown in Figure 2b, after being saturated in the opposite direction in the same manner by tilting the sample by −25°. Ignoring the signal in the hole that can be attributed to a detector offset and stray field contributions from the hole edges, domains of different color can be clearly identified around the edges of the defect. In particular, domain walls seem to nucleate from the corners, which is expected for defects of this scale. Looking at the color wheel, the domain colors are flipped by 180° after re-magnetizing the sample in the opposite direction, which is a clear indication for the magnetic origin of the signal. Furthermore, a Lorentz microscopy image of a similar defect shows an almost identical domain wall pattern around the edges, indicated by the red arrows in Figure 2c. Therefore, DPC magnetic imaging is indeed a useful tool to investigate nano-patterned magnetic thin-film membranes.

To gauge how a periodic hole pattern affects the magnetic structure of the sample, a magnetic reversal loop for both an unpatterned Co_70_Fe_30_ film, as well as a periodic pattern of circular holes using the SEM measurements from Figure 1, were created in MicroMagus14. Thus, the results of the simulation are expected to be a good representation of the experiment. Figure 3 compares snapshots of the magnetic order in these films at (a) 25 mT where the continuous thin film is still saturated, and (b) at −1 mT close to the coercive field H_C_ of the unpatterned foil.

At 25 mT (Figure 3a), the film with the periodic hole pattern already exhibits a complex pattern of magnetic domains, whereas the unpatterned film remains in saturation. Theoretical considerations by Kronmüller [20,21] indicate that voids or non-magnetic grains in a magnetic matrix can be treated as if they had an intrinsic magnetic moment opposing the magnetization of the surrounding matrix. Additionally, defects act as nucleation and pinning centers for domain walls [3]. Therefore, the results can be interpreted such that the periodic defect grid stabilizes a regular domain pattern on the bridges between the holes by dipole interactions between the voids. It is particularly interesting that some of these stabilized domains (green) are perpendicular to the direction of the external field (red).

Figure 3b shows simulations for both films at µ_0_H_ext_ = −1 mT close to the switching field H_C_ of the unpatterned film on the left. The large vertical domains in the unpatterned film are a result of the periodic boundary conditions used for the calculation. In the film with the periodic hole grid, the size of individual domains has expanded compared to the previous snapshot at 25 mT. However, the domain distribution remains otherwise mostly unchanged, which hints at a high stability of the magnetic order in nano-patterned thin films. Even with a reversed external field, some domains remain perpendicular to µ_0_H_ext_. This is also true for the rectangular pattern, not shown in Figure 3. In this case, the orientation of the rectangles respective to the external field (i.e., the easy axis of the dipole moment) also plays a role in the resulting domain pattern. Similar studies with good agreement of the magnetic domain structure in ferromagnetic nano-patterns can also be found in other systems [22,23].

Finally, comparing the measured DPC images for both a circular (Figure 4a) and a rectangular (Figure 4c) hole grid at remanence with the respective simulated magnetic structure (Figure 4b,d), it is evident that micromagnetic simulations are indeed a powerful tool to predict the experimental results and therefore serve as a useful guide for the interpretation of such images. Measurements were again performed at remanence and at 0° tilt respective to the incident beam after saturating the sample once in the magnetic field of the objective lens at a tilt of 25°. For both cases, the measured and predicted domain patterns are in excellent agreement. With few exceptions, likely arising from the periodic boundary conditions used for the calculation, the predicted tube-shaped magnetic domains between individual holes on the grid match the experimental observation in the microscope. It is worth noting that due to the sample not being able to rotate freely in the microscope, the axes of the hole grid and the axis of the external magnetic field are not completely identical and likely off by a few degrees. Nevertheless, the predicted domains that are stabilized at an angle regarding the external field can be seen in the measured images for both samples.

Although the detailed domain patterns in Figure 3 and Figure 4 agree very well with the micromagnetic calculations and reflect the geometry of the holes, broad magnetic domains are still visible in the calculations that do not appear in the experiment. It can be assumed that this is due to the periodic boundary conditions chosen for the simulations or to the external magnetic fields experimentally accessible in the TEM. The clear answer to this discrepancy must be left to future investigations with a TEM mount capable of directly measuring the magnetic fields prevailing at the sample location.

From the simulations and experimental observations, it can be concluded that magnetic thin-film membranes with periodic hole patterns exhibit a complex magnetic domain structure that can be predicted using micromagnetic simulations. These calculations can be used as a framework for the interpretation of differential phase contrast microscopy data in the context of magnetic imaging. This also hints at an interesting magnetic switching behavior of such samples that may be worth exploring in the context of magnetoresistive systems such as GMR devices. Conversely, the simulations presented in this work can be used to explore potentially interesting pattern shapes for such an application without having to use a large amount of machine time on prototype fabrication.

## 4. Conclusions

To summarize our results, we have demonstrated a high-precision fabrication technique using focused ion beam milling to create nano-patterned magnetic membranes on SiN substrates. The process is repeatable and can produce arbitrary patterns. These samples were used for DPC magnetic imaging in the TEM. By re-magnetizing and imaging the samples several times, the magnetic origin of the observed DPC signal was confirmed.

Micromagnetic simulations using the MicroMagus 7.1. ext14 code package were successfully used to recreate the experimentally observed data and were therefore established as a framework for the interpretation of differential phase contrast images from magnetic samples. Because of the excellent agreement of measured data and micromagnetic simulations, these calculations can also be used to quickly identify potentially interesting sample structures for future experiments, which helps to reduce wasted samples and machine time.

Furthermore, the experiments revealed regular magnetic domain structures in the magnetic thin-film membranes with a periodic hole pattern, which hint at unique magnetic switching behavior. Therefore, this type of sample might be worth exploring in the context of magnetoresistive sensor applications such as GMR type devices in the future.

## Figures and Tables

**Figure 1 nanomaterials-14-00116-f001:**
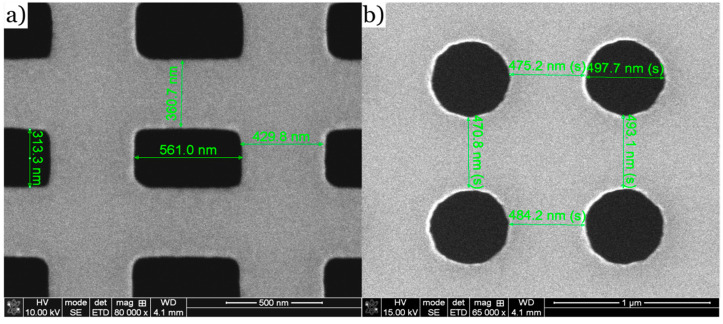
SEM images of the two geometries of the regular hole pattern. (**a**) Regular hole pattern of 560 nm × 310 nm rectangles with a lattice constant of roughly 1 µm in the x-direction and 650 nm in the y-direction. (**b**) Regular hole pattern of 500 nm holes with a lattice constant of about 1 µm in the x- and y-directions.

**Figure 2 nanomaterials-14-00116-f002:**
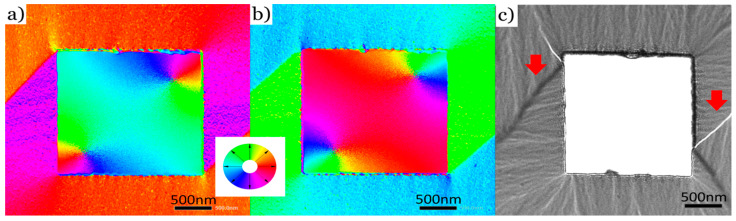
(**a**) DPC image of a 2 µm × 2 µm square defect in a 60 nm thick Co_70_Fe_30_ membrane after saturating the sample using the magnetic field of the objective lens at a sample tilt of +25°. The measurement was performed at 0° tilt relative to the incident beam. (**b**) DPC image of the same defect as in (**a**) after reversing the magnetization using the objective lens of the TEM. The colors in (**a**) and (**b**) are coded directions of magnetization and are defined according to the color wheel. (**c**) Lorentz microscopy image of a similar defect of the same dimensions. Relevant visible domain walls are indicated by red arrows.

**Figure 3 nanomaterials-14-00116-f003:**
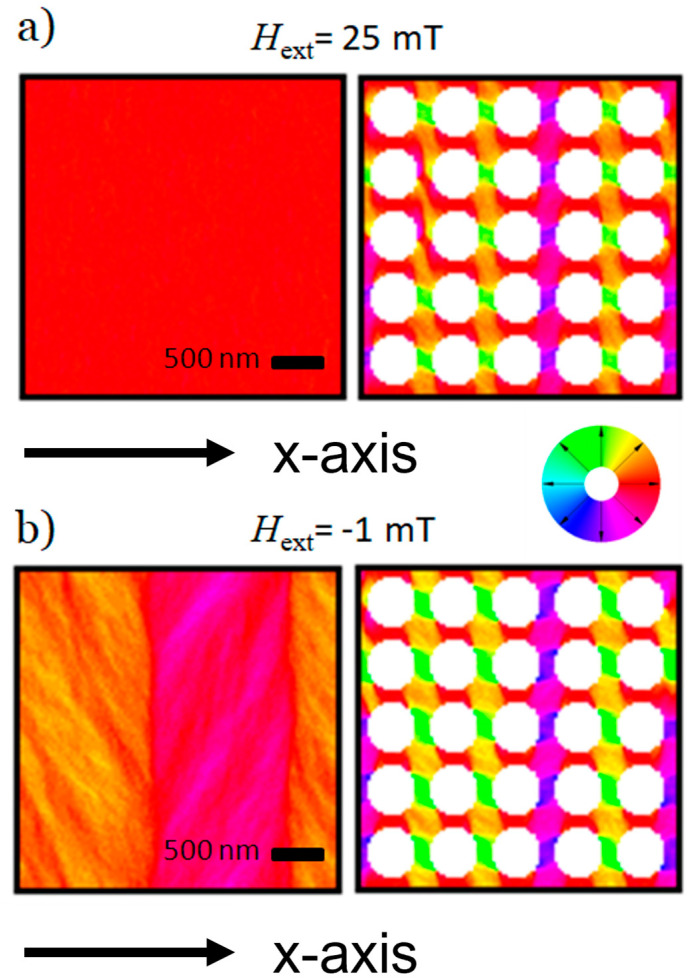
(**a**) Comparison of an unpatterned 60 nm thick Co_70_Fe_30_ membrane (left) and a Co_70_Fe_30_ membrane with a regular pattern of circular holes (right) at an external field of 25 mT parallel to the x-axis. Measurements for the circular hole pattern were taken from experimental SEM results. (**b**) A comparison of the same films at an external field of −1 mT parallel to the x-axis. Scale bars are the same for left- and right-hand side images. The colors in (**a**) and (**b**) are coded directions of magnetization and are defined according to the color wheel.

**Figure 4 nanomaterials-14-00116-f004:**
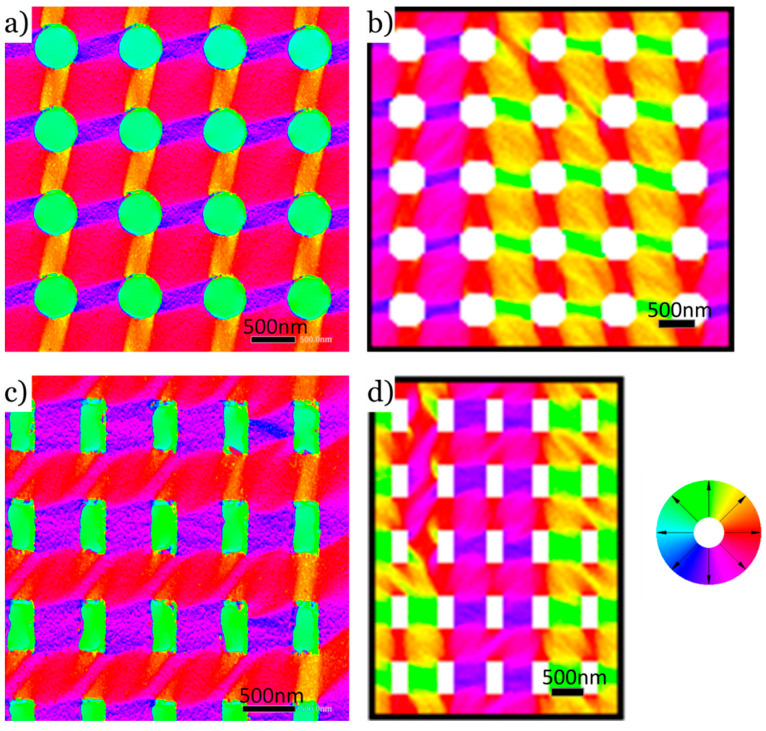
(**a**) Measured DPC image for the circular hole pattern from Figure 1 at remanence and at 0° tilt with regard to the incident electron beam. Before the measurement, the magnetization was saturated using the field of the objective lens at a tilt angle of 25°. (**b**) Simulated magnetic domain configuration at remanence for the circular hole pattern shown in Figure 1b. To properly represent the experiment, measurements from the SEM were used for the calculation. (**c**) Measured DPC image for the rectangular hole pattern from Figure 1 using the same imaging conditions as (**a**). (**d**) Simulated magnetic domain configuration at remanence for the rectangular hole pattern shown in Figure 1a. SEM size measurements were used for the calculation. The colors in (**a**–**d**) are coded directions of magnetization and are defined according to the color wheel.

**Table 1 nanomaterials-14-00116-t001:** Overview of used simulation parameters in MicroMagus 7.1ext14. Units are given to the program in the CGS unit system. To discuss the results, output values are converted to SI.

Parameter	Value	Unit
Co_70_Fe_30_:		
Layer thickness	9	nm
Saturation magnetization	1819	kA/m
Exchange stiffness constant	28.3	pJ/m
Anisotropy constant	47.3	kJ/m^3^
Py:		
Layer thickness	0.75	nm
Saturation magnetization	827	kA/m
Exchange stiffness constant	16	pJ/m
Anisotropy constant	1	kJ/m^3^
Average crystallite size (all)	20	nm
Random crystal grains (all)	yes	—
Exchange weakening at grain boundaries (all)	no	—

## Data Availability

The data presented in this study are available from the corresponding author upon reasonable request.

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
