# Peer review of "Simulated Guidance in Interpreting Nano-Patterned Co70Fe30 Film Imaging with Differential Phase Contrast"

_nanomaterials, 2024, doi:10.3390/nano14010116_

Round 1
Reviewer 1 Report
Comments and Suggestions for Authors
In this work, the authors prepared nano-patterned-milled CoFe membranes, they investigated it by TEM. This work used DPC magnetic imaging along with a simulation framework based on micromagnetic calculations, providing the interpretation of the experimental data. They found that the simulations not only match the experimental data well, but also consistent with possible structures. The authors observed magnetic microstructure of the nano-patterned magnetic membranes hints at an interesting switching behavior in an external field. In a word, this study made important contributions to nanomaterials and magnetic materials. There are some issues which should be addressed before being published.
(1) Some texts in the figures are not clear, for example, the texts of “Ni” and “Sn” et. al the STEM-EDS mapping Figure 1.
(2) The different ratio of Co/Fe affect the materials property ?
(3) Some mistakes should be revised in the references.
This reviewer suggests the acceptance of this manuscript after some minor revisions.
Comments on the Quality of English LanguageThis work can be accepted after minor revision.
Reviewer 2 Report
Comments and Suggestions for Authors
The current study in title “Simulated Guidance in Interpreting Nano-Patterned CoFe Film Imaging with Differential Phase Contrast”. The authors shows the possibility of using micro-magnetic simulation to confirm DPF magnetic imaging with TEM. I find the methodology and the idea is very interesting and will be very useful for researchers in nanomagnetism and its related phenomena. The manuscript is well-written and the presentation of the outcome results is good. Thus, I recommend accepting the manuscript after the authors fixed these points;
1- The introduction needs to enhance a little bit with more recently works in nanomagnetism and the focusing in the necessary of the suggested model (Simulation + DPF + TEM analysis). The authors have to illustrate the importance of this method and make a critical study with other methods. The introduction need to enhance.
2- The authors said; “Commonly used methods like the magneto-optical Kerr effect (PMOKE) have a limited resolution”. I think use “MOKE” instead of “PMOKE” or may the authors need to describe Polar-MOKE. Please, correct it.
3- The author said; “This technique allows us to study both electronic and magnetic properties of a sample by detecting the resulting small deflections of the transmitted electron beam using a segmented detector”. Please, add recently references support your idea.
4- The authors said; “Recent advances in DPC microscopy allowed the investigation of the electronic structure of a material with atomic resolution [9].”Please, update this sentence with more recent articles.
5- The authors should add reasons for selecting (nano-patterned Co70Fe30 membranes). Normally for the primary investigations the scientists use will know magnetic materials for the micromagnetic modeling to avoid any additional parameters can be affected on the results such as Py or Fe, Co,….
6- The authors should add paragraph in the introduction describe the reasons, advantage and facilities can be obtained for using (MicroMagus 7.1ext code package) and compare it with other well known software’s are commonly used in micromagnetic simulation such as Mumax, OOMMF,….
7- The sample preparation needs to be enhanced with more information about the target preparation. In addition, the thin film fabrication missing a lot of information such as deposition parameters, whether the samples rotate or not, applying a magnetic field to control the magnetic anisotropy of the samples, the distance between the target and the magnetron sputtering, which kind of sputtering technique, RF, DC, AC and so on.
8- The authors said;” This ensures that the in-plane component of the up to 2 T field of the objective lens saturates the magnetization of the sample and therefore induces a preferred direction for the magnetic domains close to remanence.” Could please add more dials and explanations for this specific configurations.
9- The differential phase contrast measurements in the experimental part is so confusing. If possible add illustrative figure to summarize the measurement this will facility a lot for non-expert readers.
10- It will be great if the authors add the measured M-H loops and compare with the simulated loops for both direction (easy and hard axis).
11- The magnetic behaviour plotted in Figure 3 is commonly obtained for such ferromagnetic Nano-patterned thin films with square lattice. The authors should mentined to more recently studied matches with magnetic domain structure of ferromagnetic Nano-patterned thin films.
Reviewer 3 Report
Comments and Suggestions for Authors
This manuscript described an agreement between the experimental and simulation of micromagnetic domain patterns on nano-patterned magnetic membranes on SiN substrates fabricated by focusing ion milling. The experimental magnetic domains were revealed using the Differential Phase Contrast (DPC) magnetic imaging in the TEM. The micromagnetic simulations using the MicroMagus 7.1. ext14 code package were success fully used to recreate the experimentally observed data. This is a piece of good work; however, it is not a very novelty job. The micromagnetic simulation study is quite a mature calculation method. Focusing ion milling to fabricate the nano-patterns on magnetic membranes is not a new technology also. The DPC technique was also developed more than 10 years ago. However, most of the past micromagnetic study relies on the tool of magnetic force microscopy (MFM) in the past, while, in this work, DPC of STEM is used successfully. It can be published in Nanomaterials if the following questions can be elaborated.
A few questions:
1. Despite the agreement between DPC image and micromagnetic simulation, still some discrepancies between them. More interpretation of those discrepancies can be made.
2. In table 1, the parameters used as input to MicroMagus 7.1 code. Are those parameters were adjusted to fit the image pattern or they are fixed to do the calculation only.
3. In table 1, the anisotropy constant of Py is far different to CoFe. Are those parameters close to the real thin Py and CoFe films experimentally?
4. In the Fig. 2, the magnetic moment is indicated by red arrows. But, no “red” arrow in Fig. 2.
5. In Fig. 3, the applied field parallel to X-axis, but the X-axis is never defined in the main text.
6. In line 207: Should the Fig 4b be Fig. 4c?
7. In line 209: Should the Fig. 4c, 4d be Fig. 4b, 4d?
Reviewer 4 Report
Comments and Suggestions for Authors
This paper introduces a simulation-based framework designed to interpret DPC magnetic imaging within TEM.
I would recommend it for acceptance after the minor points listed below are addressed.
1. In page 2, line 56, the authors misspelled "sputter" as "putter".
2. In Figure 2 c), red arrows are not indicated.
3. In page 6, line 208, "Fig. 4b" should be "Fig. 4c", and in line 509, "Fig. 4c" should be "Fig. 4b".
Round 2
Reviewer 2 Report
Comments and Suggestions for Authors
The authors fixed all of my comments and the current version of the manuscript is suitable for publication in the present form.